# Methods for Stratification and Validation Cohorts: A Scoping Review

**DOI:** 10.3390/jpm12050688

**Published:** 2022-04-26

**Authors:** Teresa Torres Moral, Albert Sanchez-Niubo, Anna Monistrol-Mula, Chiara Gerardi, Rita Banzi, Paula Garcia, Jacques Demotes-Mainard, Josep Maria Haro

**Affiliations:** 1Research and Development Unit, Parc Sanitari Sant Joan de Déu, 08830 Barcelona, Spain; teretmoral@gmail.com (T.T.M.); anna.monistrol@sjd.es (A.M.-M.); josepmaria.haro@sjd.es (J.M.H.); 2Melanoma Unit, Dermatology Department, August Pi i Sunyer Biomedical Research Institute (IDIBAPS) and Hospital Clínic, 08036 Barcelona, Spain; 3Center for Networked Biomedical Research on Rare Diseases (CIBERER), Carlos III Health Institute, 28029 Madrid, Spain; 4Center for Networked Biomedical Research on Mental Health (CIBERSAM), Carlos III Health Institute, 28029 Madrid, Spain; 5Department of Social Psychology and Quantitative Psychology, University of Barcelona, 08028 Barcelona, Spain; 6Centre for Health Regulatory Policies, Istituto di Ricerche Farmacologiche Mario Negri IRCCS, 20156 Milan, Italy; chiara.gerardi@marionegri.it (C.G.); rita.banzi@marionegri.it (R.B.); 7ECRIN, European Clinical Research Infrastructure Network, 75013 Paris, France; paula.garcia@ecrin.org (P.G.); jacques.demotes@ecrin.org (J.D.-M.)

**Keywords:** cohorts, sample size, stratification, personalized medicine

## Abstract

Personalized medicine requires large cohorts for patient stratification and validation of patient clustering. However, standards and harmonized practices on the methods and tools to be used for the design and management of cohorts in personalized medicine remain to be defined. This study aims to describe the current state-of-the-art in this area. A scoping review was conducted searching in PubMed, EMBASE, Web of Science, Psycinfo and Cochrane Library for reviews about tools and methods related to cohorts used in personalized medicine. The search focused on cancer, stroke and Alzheimer’s disease and was limited to reports in English, French, German, Italian and Spanish published from 2005 to April 2020. The screening process was reported through a PRISMA flowchart. Fifty reviews were included, mostly including information about how data were generated (25/50) and about tools used for data management and analysis (24/50). No direct information was found about the quality of data and the requirements to monitor associated clinical data. A scarcity of information and standards was found in specific areas such as sample size calculation. With this information, comprehensive guidelines could be developed in the future to improve the reproducibility and robustness in the design and management of cohorts in personalized medicine studies.

## 1. Introduction

A personalized medical approach requires patient stratification, leading to identifying homogeneous patient clusters relevant for diagnosis and treatment. Identification of patient subgroups based on a limited number of determinants (companion diagnostics) is gradually being replaced by patient stratification based on complex, multimodal profiling using biological (genomic, epigenomic, transcriptomic, proteomic, metabolomic, etc.), clinical, imaging, environmental (microbiome, exposome) and real-world data (wearable sensors and lifestyle data) [1,2]. Data analysis through conventional methods or machine learning algorithms establishing complex correlations leads to data-driven patient stratification, which can be independent of understanding the disease mechanism. Thus, for this initial phase in the context of personalized medicine, the construction of large cohorts is essential to stratify homogeneous patient groups adequately. This stratification can be used to define a new disease taxonomy, refine diagnostic procedures or propose more specific treatments for individual patient groups. In addition, a proper validation cohort can assess the reproducibility, robustness and validity of the grouping in another sufficiently large sample of patients [3,4,5,6].

Patient clusters are helpful in clinical practice if they are adequately defined through exploratory studies. The criteria to determine the patient strata are suggested, and enough evidence for their use in patient care is obtained through validation studies. Accordingly, it is essential to have appropriate methods and tools for the adequate design of stratification and validation cohorts and the resulting clusters. We have conducted a scoping review to identify the state-of-the-art and gaps of methods and tools involved in the design, management and validation of cohorts used in establishing patient clusters. 

This scoping review is part of the PERMIT project (Personalised Medicine Trials), aimed at mapping the methods for personalized medicine (Box 1) research and building recommendations on robustness and reproducibility of different stages of the development programs. While several categorizations may be proposed, the PERMIT project considers four main building blocks of the personalized medicine research pipeline: (1) Design, building and management of stratification and validation cohorts; (2) application of machine learning methods for patient stratification; (3) use of preclinical methods for translational development, including the use of preclinical models used to assign treatments to patient clusters; (4) evaluation of treatments in randomized clinical trials. This scoping review covers the first building block in this framework. We considered the scoping review approach to be the most suitable tool to respond to the broad ambit of these objectives. Compared to systematic reviews that aim to answer specific questions, scoping reviews are used to present a comprehensive overview of the evidence about a topic. They are helpful to examine emerging areas to clarify key concepts and identify gaps [7,8].

Box 1.What is Personalised Medicine?According to the European Council Conclusion on personalised medicine for patients personalised medicine is a medical model using characterisation of individuals’ phenotypes and genotypes (e.g. molecular profiling, medical imaging, lifestyle data) for tailoring the right therapeutic strategy for the right person at the right time, and/or to determine the predisposition to disease and/or to deliver timely and targeted prevention [9]. In the context of the Permit project, we applied the following common operational definition of personalised medicine research: a set of comprehensive methods, (methodological, statistical, validation or technologies) to be applied in the different phases of the development of a personalised approach to treatment, diagnosis, prognosis, or risk prediction. Ideally, robust and reproducible methods should cover all the steps between the generation of the hypothesis (e.g., a given stratum of patients could better respond to a treatment), its validation and pre-clinical development, and up to the definition of its value in a clinical setting. (ref protocol Zenodo) [10].

## 2. Materials and Methods

We followed the methodological framework suggested by the Joanna Briggs Institute [11]. The framework consists of six stages: (1) Identifying the research questions, (2) identifying relevant studies, (3) study selection, (4) charting the data, (5) collating, summarizing and reporting results and (6) consultation.

We published the study protocol in Zenodo before conducting the review [10]. Due to the iterative nature of scoping reviews, deviations from the protocol are expected and duly reported when they occurred. We used the PRISMA-ScR (Preferred Reporting Items for Systematic reviews and Meta-Analyses extension for Scoping Reviews) checklist to report our results [12].

### 2.1. Research Questions

The scope of this review on design, validation and management of cohorts has been structured through the following research questions:What are the differences, pros and cons of the prospective and retrospective design of stratification and validation cohorts?Which are the methods for defining the optimal size of stratification/validation cohorts? What are the prerequisites and methods used for the integration of multiple retrospective cohorts?What type of data (omics, imaging, exposome, lifestyle, etc.) are included and how are data generated?What are the tools used in personalized medicine for data management and multimodal data analysis?What quality of cohort data is needed to obtain a biomarker or multimodal data profiling? Are there requirements to monitor the collection of associated clinical data?Which current and reliable designs exist for the stratification (or clustering) in personalized medicine?Which methods and tools are used to build the cohorts for validation of patient strata?What are the methods for the evaluation of the risk of bias?What is the outlook of data generation seen as (CE-labelled) in vitro diagnostics?

We focused on three case models: Oncology, Alzheimer’s disease (AD) and stroke. We chose these three fields because they are in different development phases of personalized medicine, which allows us to explore methods and strategies in the different stages of progress. These fields also use a different kind of data to stratify patients. Oncology is where personalized medicine was first applied, and targeted therapies and diagnostics have been mainly focused on. Moreover, several applications of biomarkers for the successful stratification of patients with a given type of cancer exist, most of them based on molecular data, especially genomics [13,14]. AD research in personalized therapies and diagnostics nowadays is showing its firsts results, based on imaging, cognitive and molecular data [15]. Stroke is currently opening to personalized medicine, with some approaches and studies in more initial phases. Most of the data used for patient stratification are imaging and molecular data [16].

### 2.2. Information Sources and Search of Studies

We limited our search from 2005 to April 2020 and restricted it to English, French, German, Italian and Spanish. Relevant studies and documents were identified balancing feasibility with breadth and comprehensiveness of searches. We searched scientific manuscripts of reviews in the fields of cancer, stroke and AD in PubMed, EMBASE, Web of Science, Psycinfo and the Cochrane Library (search date: March–June 2020) to answer the research questions. We used keywords for illnesses, (e.g., “dementia”, “Alzheimer”, “neoplasms”, “cancer”, “cerebrovascular disorders” or “ischemic attack, transient”), for design (e.g., “validation studies” “cohorts design”, “prospective cohort” or “retrospective cohort”), for methods (e.g., “methods”, “data integration” or “bias”) and for personalized medicine (e.g., “stratified medicine”, “biomarker”, “precision medicine” or “personalised medicine”). We also searched for grey literature on relevant websites. Appendix A reports the search strategies applied. 

### 2.3. Selection of Sources of Evidence

We exported the references retrieved from the searches into the Rayyan online tool [17]. Duplicates were removed automatically using the reference manager Endnote X9 (Clarivate Analytics, Philadelphia, PA, USA) and manually verified. Two reviewers (TTM, AMM) screened all titles and abstracts of the retrieved citations independently. Papers having a different study design to those in the inclusion criteria, focusing on other illnesses or clearly out of our aim, were discarded. In cases of disagreement, a consensus was determined by discussion. After this selection, a full-text review was conducted by one of the investigators (TTM) to select the documents included in the review. During this phase, we identified several papers not solely addressing one of the three case models foreseen in the protocol, but including at least one of them. They reported information that was deemed relevant to the aim of the review. Therefore, the authors agreed on their inclusion and referred to it as “multiple disease reviews”. This inclusion was a deviation from the protocol.

### 2.4. Data Charting Process

Information was summarized in tables by one reviewer (TTM) and checked by a second reviewer (ASN) to ensure data quality. In cases of disagreement, consensus was obtained through discussion. It was not within the remit of this scoping review to assess the methodological quality of individual studies included in the analysis.

### 2.5. Consultation Exercise 

The members of the PERMIT consortium, associated partners and the PERMIT project Scientific Advisory Board discussed the preliminary findings of the scoping review in a two-hour online workshop. 

## 3. Results

In total, the database searches retrieved 2362 records. Nine additional records were retrieved through the included reviews and grey literature. After the screening process, 50 reviews and grey literature documents about European regulation were included: Nineteen on oncology, two on AD, two on stroke and 27 multiple disease reviews. Figure 1 reports the PRISMA flowchart for article selection. More information about the included reviews and grey literature is provided, respectively, in Appendix A. The answers to each of the research questions raised are presented in the following subsections.

### 3.1. What Are the Differences, Pros and Cons of the Prospective and Retrospective Design of Stratification and Validation Cohorts?

Fourteen reviews addressed this question. We found that the cohorts identified were primarily prospective [18,19,20,21,22,23,24,25,26,27,28,29,30,31]. Prospective cohorts were predominantly used because they enable the optimal measurement of predictors and outcomes, the risk of recall bias is reduced and causality is easier to establish. On the other hand, the main advantage of the retrospective design is that no follow-up of participants is required. Hence, costs are lower. In addition, the use of retrospective cohorts facilitates the study of rare diseases. Further information about the pros and cons of these study designs is provided in Appendix A. Additionally, Appendix A shows different specific subtypes of prospective designs found in oncology studies with their pros and cons. 

### 3.2. Which Are the Methods for Defining the Optimal Size of Stratification/Validation Cohorts? What Are the Prerequisites and Methods Used for the Integration of Multiple Retrospective Cohorts?

Five reviews focused on this question. We have not found generic methods for estimating the optimal size of stratification and validation cohorts, but we have retrieved different strategies to determine the size of the cohorts [21,22,23,32,33]. 

The first strategy is using pre-selected fixed numbers based on rule-of-thumb approaches. For example, in retrospective studies in oncology, it has been suggested that testing samples should include approximately 110 subjects without cancer and 70 subjects with cancer, and training samples should include at least the same size. In prospective studies the sample size recommended is larger. Moreover, we found that to validate a prediction model with a binary outcome, a minimum effective sample size of 100 participants with and 100 without the outcome event is needed. Furthermore, for each study of a candidate predictor, at least ten events are required for categorical outcomes [21,22,23].

Another strategy to determine the appropriate cohort size considers certain aspects, such as the type of study, the disease of interest and the specific objectives and the statistical criteria. A review on AD suggested the following aspects to take into account when calculating the optimal size: The use of two-stage “summary measures” analyses or analysis based on longitudinal parametric models; the effect size; the type of target population; and the possibility of conducting a test analysis on pilot data [32].

Given that large sample sizes may be required, one review proposed integrating multiple retrospective cohorts to obtain a sufficient sample size to achieve the necessary statistical power for data analyses [33]. This review focused on microarray data integration and proposed two analysis strategies: One approach consisted of pooling different microarray experiments to form a single dataset which allowed easy clustering or intersection operations (data aggregation), and the other approach was to analyze each microarray experiment and then aggregate the statistical results of all separate experiments using meta-analytic techniques. Some of the methods mentioned are the Mantel–Haenszel method, meta-regression or the latent variable approach. Moreover, the review mentioned methods to explore the relationship of the study characteristics, making the integration of cohorts easier. Finally, this study also highlighted the challenge of ethics and infrastructure for managing patient data integration. Ideally, it requires institutional commitment to building a comprehensive and global infrastructure. Still, because global infrastructures are expensive, the review suggests that a local one could be developed, which would involve considerably less money and time [33].

Despite the advantages of integrating multiple retrospective cohorts to achieve larger sample sizes, we did not find more reviews on this topic.

### 3.3. What Type of Data (Omics, Imaging, Exposome, Lifestyle, Etc.) Is Included and How Are Data Generated?

We found 25 reviews providing information about this topic. Most of the reviewed studies used multimodal data (see Table 1). Each type of data has its advantages and disadvantages concerning its costs, the data it provides or the difficulty of being quantified [18,20,21,22,23,24,28,32,33,34,35,36,37,38,39,40,41,42,43,44,45,46,47,48,49]. 

Some examples of the type of data and their pros and cons are:Genetic variation data. This type of data is an unbiased source of the genetic basis of disease and allows the direct inference of causality.Epigenetics data. It is useful to know the functional impact and makes the inference of causality easy, but it does not apply to all phenotypes.Gene expression data. It shows the picture of an intermediate step towards the phenotype.Proteomics and metabolomics data. This type of data is the closest to the phenotype, but both data are expensive, especially proteomics.Microbiome data. It is very close to the phenotype and measures a combination of genetic and environmental influences. The combination of genetic and environmental influences makes it difficult to infer the direction of causality.Image and environmental data on different biological layers. These two types of data are hard to measure quantitatively [34].

The aggregation of different type of data has also been studied. Radiogenomic studies offer the opportunity to assess the relationships between genotype features, intermediate phenotype features, radiomic features and phenotypic clinical outcomes [37]. One review highlighted that the association between the functional properties of the microbiome and other features could be helpful to understand the phenotype of the different illnesses better [36]. Thus, that review found the development of tools and algorithms to integrate 16S sequence information and metagenomics with metatranscriptomics and metaepigenomics relevant [36].

### 3.4. What Are the Tools Used in Personalized Medicine for Data Management and Multimodal Data Analysis? 

Data management and the analysis of data are key factors to make the most of the data. Therefore a large number of reviews, a total of 24, provided information on the methods and tools used for this purpose [18,22,24,33,34,37,38,39,41,43,44,45,46,48,50,51,52,53,54,55,56,57,58,59]. An outline of methods and tools has been collected in Appendix B. Methods and tools for data analysis and methods for different types of data integration are reported separately below. 

#### 3.4.1. Methods and Tools for Data Management and Multi-Omics Data Analysis 

Regarding data management tools in oncology, some data warehouses, such as Data Warehouse for Translational Research, enable users to interrogate clinico-pathologic data in a multidimensional manner dynamically [37]. The Joint Modelling approach was one of the methods found for analysis, which recommends integrating both large-scale omics and non-omics data [60]. Other methods involved network analysis, such as Multiblock Partial Least Squares (or projection to latent structures, MBPLS). This method seeks to maximize covariance between summary vectors derived from each omics data block and a biological phenotype or response [34].

One review based on multiple diseases analyzed the tools for an integrative approach using multi-omics data as repositories or visualization portals and the challenges in integrating data sets [20]. A standard classification for integrative multi-omics methods has not been established due to three reasons: The same methods can achieve different objectives; pipelines presented for addressing a particular problem can also be used to solve another problem (e.g., other types of omics), and several tools can be used in a supervised or unsupervised setting [56]. Some reviews also pointed out that uniformity in the process and analysis of multi-omics data may lead to a reduction in spurious signal (caused by technical limitations of platforms) and an increase in the identification of associations. The main difficulty in generating uniform strategies is the inability to know what the models should include and, therefore, the most effective modelling strategies [18,41].

#### 3.4.2. Methods for Data Integration

One oncology review pointed out that integrative data sets often do not have a standard format, highlighting the importance of building an infrastructure to house and manage these data. Further work is needed to create an infrastructure for storing genetic and transcriptomic data in Electronic Health Records (EHR) [38]. Moreover, there is a need to determine which information will be reported back to a patient and incorporated into an EHR, which will require concerted efforts from clinicians and researchers [38]. Despite this, one review described some strategies for coupling EHRs to genomic datasets to determine genotype–phenotype associations [60].

Constraint-based modelling methods are relevant for integrating different omics layers at genome-scale, which provide a mechanistic link between the genotype and their phenotypic observables [43]. These integration methods can be “vertical”, including omics layers, or “horizontal”, based on the same model but focusing on different environments (cancers or growth conditions) [61,62]. Moreover, one review focusing on multiple diseases classified the integration methods by taking into account the features that these methods help to improve: Velocity (e.g., anytime algorithms) or variety (e.g., matrix factor-based methods, or Graph-regularized non-negative matrix tri-factorization). It also pointed out that integrating many data types, including exposomic and metagenomic data, is a focus for future studies [51].

Regarding the tools for data integration, these mainly concentrate on machine learning and deep learning-based approaches. These approaches allow the computational flexibility to effectively model and integrate almost any omics type [44,55].

Finally, all reviews indicated that the existing statistical methods mainly address data analysis and management. Multi-layer data integrative methods for multi-omics-derived data need to be developed and improved [45].

### 3.5. What Quality of Cohort Data Is Needed to Obtain a Biomarker or Multimodal Data Profiling? Are There Requirements to Monitor the Collection of Associated Clinical Data? 

Despite the relevance of the quality of data and the monitoring of data collection for the reliability of the results, we have not found specific information about the quality of data in the cohort building step. Furthermore, we have also noticed an absence of information regarding the monitoring of associated clinical data. The closest information found was related to guidelines regarding sample size [21,22,23] or to requirements for statistical analyses [32,33], described in other sections of this article.

### 3.6. Which Current and Reliable Designs Exist for the Stratification (or Clustering) in Personalized Medicine? 

Four reviews addressed this question. Reliable clustering methods are needed to develop accurate studies of personalized medicine. Hence information about stratification methods and their reliability are relevant [35,47,55,63].

#### 3.6.1. Oncology

One review pointed out that staging systems for hepatocellular carcinoma should be considered important for treatment recommendations, but current methods are not optimal. Novel methods, such as those based on systems biology, pathology-specific biomarker panels and multilevel diagnostics, are ongoing and present various degrees of portability, ranging from general to particular algorithms, and allow a more accurate patient stratification [35].

#### 3.6.2. Alzheimer’s Disease

We found that pathology-specific PET-imaging data is accurate for understanding the regional distribution of molecular disease processes for patient clustering. Still, a direct combination of high-dimensional information from multiple modalities may achieve a more comprehensive definition of distinct disease subtypes in AD and related dementia. Several statistical methods for clustering have been found: Ward’s clustering; unsupervised graph-theory-based clustering approach; non-negative matrix factorization; unsupervised random-based clustering; and assigning patients to subtypes in a probabilistic fashion method [47].

#### 3.6.3. Multiple Disease Reviews

In these reviews, the methods to stratify patients were classified in the following types: One, clustering based on groups of patients with similar disease evolution; two, dimensionality reduction, that is, methods that select the representative features to characterize specific groups of patients; three, similarity methods, those that define a novel measure of similarity among patients; four, software tools, or methods based on a software solution implementing the proposed approach; and finally, a combination of clustering or similarity metrics and supervised approaches, i.e., methods where the features, that characterize a group of patients are then used to solve a classification task [55,63].

### 3.7. Which Methods and Tools Are Used to Build the Cohorts for the Validation of Patient Strata?

Eleven reviews dealt with the building of validation cohorts [20,22,25,27,35,40,49,57,63,64,65].

#### 3.7.1. Oncology

Reviews on oncology conclude that statistical methods to correct the potential effect of multiple analyses (such as Bonferroni or Hochberg’s correction) prevent the reporting of false-positive results. External validation on comparable series of patients is at best adequate but rarely possible. Alternatively, complex internal cross-validation procedures could ensure biomarker stability. The methods defined in the reviews have been schematized in Appendix A [25,27,35,64].

#### 3.7.2. Multiple Disease Reviews

One review suggested that internal validation often gives exaggerated estimates of biomarker performance, probably due to feature selection bias, population selection bias or optimization biases [22,65]. External validation is the most rigorous form of model validity assessment. However, biases may also be introduced in this type of validation and may have different levels of rigor (e.g., replication by the same team of investigators or not). Despite the importance of replication, it is important to note that most biomarker studies never have independent replication [22,40,57,65]. Moreover, another review focused on propensity score methods and methods for matching. Both choices are good for enrolling matched control groups in biomarker studies [63].

Another review focused on the different designs of pharmacogenetic studies, each design having a different structure of validation of cohorts. The aforementioned review classified the designs into targeted or enriched designs (pre-screening step in which patients are selected for the study based on genotype), stratification design (pre-screening step where all subjects are assigned to groups based on genotype and then randomly assigned to treatment) and adaptive design (enrolled patients are randomly assigned to treatment groups and treatments are compared as part of a primary study objective) [20].

#### 3.7.3. Stroke

One review concluded that both propensity score matching and one-to-one matching are suitable approaches [49]. The propensity score estimates the probability that one participant belongs to the stroke group only considering their baseline characteristics. This approach aims to match each participant with stroke with the available control with the closest propensity score. Worth noting is that a large pool of enrolled controls is required to obtain patients and controls with similar scores. In one-to-one matching, the investigator enrolls control participants that are perfect or nearly perfect matches for each stroke participant based on their demographic features and other potential confounders. This approach requires computer programming support to manage all the EHR information. According to the review, one-to-one matching is a much more targeted approach than propensity score matching [49].

### 3.8. What Are the Methods for the Evaluation of the Risk of Bias? 

Eight reviews were related to this question. They reported that most of the studies in personalized medicine do not publish a complete analysis of the risk of bias. However, it is important to evaluate and analyze the risk of bias to minimize it [18,19,24,26,27,29,42,57].

In the oncology reviews, there were two methods to avoid the bias caused by the lack of data: Inverse probability weighting and multiple imputation [24]. The conclusion reached was that there are unavoidable sources of bias, but the above approaches can reduce them. Other conclusions were that general scientific standards should be established for scientific rigor and reproducibility and that there is a need for standardization of statistical tools and methodologies in specific fields [24,29].

Multiple disease reviews reported different methods to mitigate different kinds of bias (Table 2) [18,19,42,57]. Some reviews suggested that the main problem is the unavoidable errors in fundamental data, which can be mitigated using extensive cohort studies. Separating bench researchers and clinical epidemiologists is a suitable approach to cover both kinds of research [42]. Another bias with a non-statistical origin is the “publication bias”, which occurs because research groups tend to report only the best results. Publishing positive and negative outcomes is a need to reduce this kind of bias [18]. Another review noted the relevance of detecting the type of bias, analyzing its possible cause and devising a strategy to address it [19].

Finally, some reviews on oncology focused on how to measure the risk of bias. The tool Quality in Prognosis Studies (QUIPS) assesses the risk of bias in six domains: Study participation, study attrition, prognostic factor measurement, outcome measurement, study confounding and statistical analysis and reporting. In addition, the reviews concluded that, although there is a black box in the analysis of the risk of bias, three sources of risk of bias are usually analyzed:Ooutcome measurement, study confounding and statistical analysis and reporting [26,27].

### 3.9. What Is the Outlook of Data Generation Seen as (CE-Labelled) In Vitro Diagnostics?

The new European regulatory framework for in vitro diagnostics will be fundamental when designing and implementing further studies and advances in personalized medicine. In vitro diagnostics are frequently used in personalized medicine to identify patients likely to benefit from specific treatments or therapies [66]. We found five grey literature documents about European regulation [66,67,68,69,70].

The new regulation for in vitro diagnostics is meant to be applied in the European Union before May 2022 [69,70]. This new regulation will likely improve the quality and effectiveness of these instruments by increasing the requirements in their development and approval, which will translate into advances in drug developmental and diagnostic tests, fostering personalized medicine. 

In 2017, 35% of all new drugs approved by the FDA were personalized, and this number will continue growing [67,68]. It is likely that in vitro diagnostics will go through an exponential increase, leading to the market authorization of new drugs that rely on them in the near future [71,72,73,74]. At the moment, some techniques such as CRISPR Technology [75], liquid biopsy [76] and epigenetics [77] are advancing particularly fast. 

## 4. Discussion

### 4.1. Summary of Evidence

We have mapped the characteristics of cohorts used in studies of personalized medicine, the methods and tools involved in the design and selection of these cohorts and the validation and analysis of data. Finally, we have examined the regulatory framework for the development of cohorts in the field of personalized medicine. Further efforts are required towards the unification and organization of the existing information to be easily used in the initial stages of personalized medicine. Regarding the cohort design, we have explored three aspects: The pros and cons of a prospective or retrospective design, the calculation of sample size and assessing the risk of bias. 

We have found that the retrospective design requires fewer resources than the prospective one, but the risk of recall bias should be carefully considered. Moreover, it has been found that the choice of study design depends not only on the study’s objective but also on its feasibility [18,19,20,21,22,23,24,25,26,27,28,29,30,31].

We have not retrieved a standardized approach to sample size calculation to be explicitly applied to cohorts in personalized medicine. This gap may be due to the complexities of estimating the sample size in analyses with widely different variables. There are, however, two different perspectives to calculate the appropriate sample size. One is to use pre-selected fixed numbers based on rule-of-thumb approaches [21,22,23], and the other is to consider certain aspects of the study, such as type of study, specific objectives or statistical criteria, to decide the appropriate sample size [32].

The type of study, the objectives, if the parameters are study-specific driven, the type of data and the choice of statistical methods should be considered for an adequate sample size calculation [33]. Other factors that may influence the sample size requirements are whether the final intervention will be pharmacological or non-pharmacological or whether the setting is supervised or unsupervised [78].

In order to build a cohort with a large sample size, a proposed helpful strategy is to integrate existing multiple retrospective cohorts. This is one of the multiple reasons why it is essential to develop strategies and tools that facilitate the integration of multiple retrospective cohorts [33]. Despite its importance, we have found a lack of reviews on this topic, suggesting a need for research to standardize the prerequisites and find new tools to integrate different cohorts. 

A narrative review about methods for harmonizing cohorts classified the advantages of data integration into three types. The first type concerns advantages related to the theoretical rationale to increase representativeness. For example, to cover different populations, to be able to know the source of heterogeneity, to understand complex illnesses through different data types better and to be able to identify the natural gaps in research. The second concerns advantages related to statistical issues, for example, that larger samples are required to obtain sufficient data for the analysis. Finally, there are the advantages related to practical issues. Obtaining existing data is an opportunity to reach results faster, compared to initiating a new study. Thus, it may be a more cost-effective and an unnecessary duplication of work or an additional burden on the target population [79].

Another helpful tool proposed in other reviews to estimate the suitable sample size for different settings is the use of pilot data simulation [80,81,82,83,84,85,86].

As with any other research projects, cohorts for stratification and clustering may be affected by various biases. Careful planning and conduction of studies can mitigate them, and there are tools available for the analysis of that risk [26,27]. Despite this, only a minority of studies conduct and publish a complete analysis of the risk of bias. Therefore, it is recommended that researchers conduct and report complete analyses of the risk of bias in their studies.

We have also explored the data used in personalized medicine studies, finding that each type of data has its particular and unshared characteristics. Technically, image and environmental data on different biological layers are the hardest to measure [34]. Conceptually, these differences make the studies with different types of data challenging, driving to new and emerging research fields [37].

We have not found specific information about the quality of cohort’s data or the requirements to monitor the collection of associated clinical data. This gap is of utmost importance because any stratification study should rely on high-quality data to be meaningful. Monitoring of data is relevant for traceability, and more research in this field is needed to understand the requirements of monitoring for the different kinds of omics data.

Considerable diversity of methods has been identified regarding the tools used for data management and multimodal data analysis. The strengths and limitations of the different methods have been meticulously detailed. To approach the choice of one method or another, it is important to consider if it is necessary to conduct an analysis or only to integrate data. Moreover, it is also important to consider other factors, such as the type of data (e.g., omics or non-omics) or the aim of the study (e.g., to find an association between omics data and the biological phenotype or to understand the mechanism of the illness). It is important to point out that integrative data sets often do not have a standard format for research use in oncology [38]. In other areas, results show a need to develop and improve multi-layer data integrative methods for multi-omics-derived data to provide tailored therapies [45].

To our knowledge, only one review covers some of the topics that we have developed here from the same angle. It is a narrative review that focuses on methods for harmonization of previous retrospective cohorts. Interestingly, it presents an in-depth analysis of the causes of the need for data integration, focusing on the advantages of heterogeneity and population representativeness, statistics and cost efficiency [79].

### 4.2. Gaps Detected in Stratification and Validation Cohorts in Personalized Medicine

The present scoping review has found a lack of standards and a shortage of information in the areas described below.

First, we have found different approaches to determine the optimal size, but they are rather generic, as rules of thumb that depend on various aspects of the target study. It could be advantageous to record and prioritize the factors to be considered to calculate the minimum necessary sample size. For this, further studies are needed to determine standard approaches as a reference.

Second, we have noticed a lack of methodological recommendations to integrate data from different cohorts and a lack of standards of homogeneity between cohorts to be integrated in the way of collecting the data. Taking this into account, it would be helpful to work on three aspects: The most relevant items to consider when collecting data, how data from different cohorts are integrated and the elements contemplated to integrate data from different cohorts. Moreover, it would be helpful to know the non-specified standards currently followed to collect data and the provider of such standards.

Third, we have detected a lack of harmonization tools for biological data and a lack of tools and methods to evaluate the similarity between cohorts. Further research could help identify tools in development for harmonizing biological data, and which approaches are currently used to evaluate the similarity between cohorts.

Fourth, no specific information about standards for minimum experimental data quality has been found, and requirements for monitoring associated clinical data were also missing. In the design of cohorts, the minimal sample size was not regulated; and during the conduction of the analysis in the biomarker or multimodal data profiling studies, no uniform requirements prevailed regarding statistics. In addition, no regulation has been found about the quality of cohorts in terms of revision or validation of data. For this reason, it is necessary to establish standards of minimum experimental data quality that allow the conduct of biomarker or multimodal data profiling. Regarding monitoring of the clinical data collection associated with biomarker studies, there is an important gap to be covered, and clear regulation on this would be widely advantageous.

### 4.3. Future Research and Guidelines Proposals

The final objective of the PERMIT project is to write guidelines and recommendations to improve research in personalized medicine. This scoping review has identified gaps and needs for standards regarding stratification and validation cohorts, and it defines the areas where these guidelines can be especially relevant. Useful guidelines would include information regarding sample size, gathering the factors to consider for its calculation. Regarding the integration of multiple retrospective cohorts, these guidelines would collect the existing methods and create methodological standards for integration. Finally, regarding quality of data, they would develop standards on data quality and requirements to monitor the collection of associated clinical data. 

### 4.4. Limitations

This review focuses on three case models: Oncology, Alzheimer Disease and stroke. These three illnesses have a significant impact on public and individual health. They are in three different phases of maturity within personalized medicine, and this allows the study of methods and strategies in different levels of development, using different kinds of data to stratify patients. However, they are not necessarily representative of the entire field of personalized medicine.

Another limitation is that, given the quantity of information and the breadth of this scoping review, the information presented was extracted only from reviews; information was not retrieved directly from the primary studies of the reviews. 

## 5. Conclusions

This scoping review collects and presents an analysis of the methods and tools currently used to design, build and analyze stratification and validation cohorts. It explores the current use of these tools and methods, the criteria to consider when applying them and the existing regulation that affects them.

Gaps and lack of uniformity have been found in sample size calculation, integration of multiple retrospective cohorts, quality of data of cohorts needed to obtain a biomarker or multimodal data profiling and the requirements to monitor the collection of associated clinical data in biomarkers studies. These gaps have led us to suggest the creation of more comprehensive and specific guidelines in the future.

## Figures and Tables

**Figure 1 jpm-12-00688-f001:**
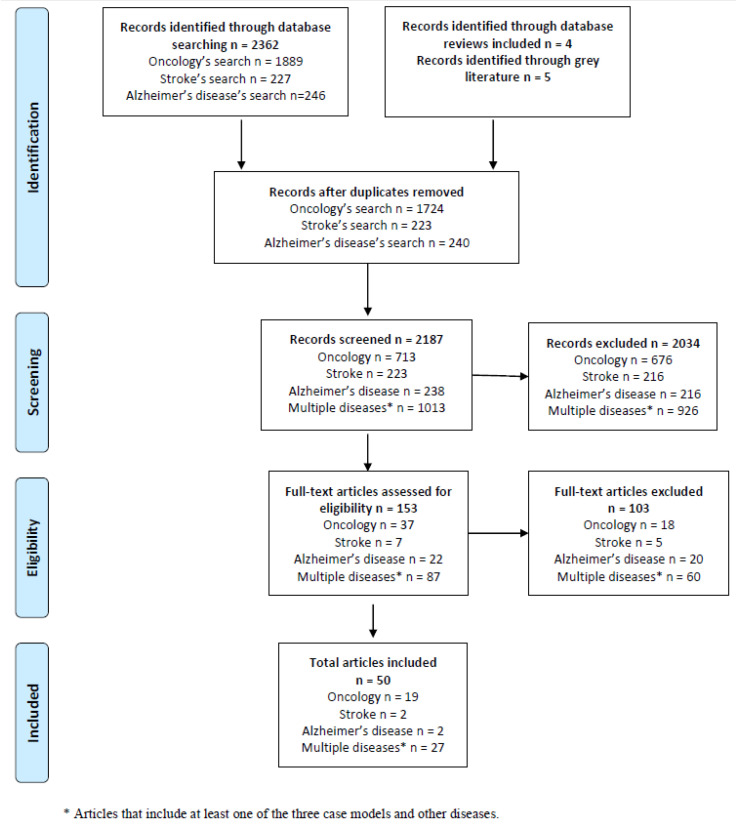
PRISMA flowchart describing process for article selection.

**Table 1 jpm-12-00688-t001:** Summary of the quantity of information found by type of data.

	Methods and Tools	Most Frequent Strategy Used
Within-Subject Correlation	To quantify intraclass correlation: Modifications of Pearson’s product-moment correlation coefficient. Comparisons based on the generalized estimating equations generated by mixed-effect models.	Analysis of data using a mixed-effect linear model that can accommodate a dependent variance-covariance structure.
Multiplicity	Controlling the family-wise error rate (Tukey, Bonferroni, Scheffe and other). Approach to controlling the false discovery rate used in biomarker studies: Benjamini and Hochber.	Analysis of data using a methodology that controls the family-wise error rate.
Multiple Clinical Endpoints	The selection of a single primary endpoint for formal statistical inference, considering that the endpoints are possibly biologically related and positively correlated. Creating a univariate outcome by combining multiple clinical endpoints (weighted measures taking into account the relevance of each endpoint). To compare the two samples based on the endpoint of highest priority first, and, if no winner can be determined, would one move to the endpoint of the next highest priority.	Analysis of data by prioritizing the relevant endpoints or by using a composite endpoint.
Selection bias	To adjust for age, stage, treatment and so forth. Matched samples.	Analysis of data using a multivariate model to simultaneously adjust for confounders Obtention of matched samples. Propensity score weighted.
Publication bias	To publish positive and negative results.	Encourage the objective assessment of molecular signatures by reporting both positive and negative outcomes. Make data publicly available after publication.

**Table 2 jpm-12-00688-t002:** A summary of methods, tools and strategies proposed to avoid different types of bias.

	Methods and Tools	Most Frequent Strategy Used
Within-Subject Correlation	To quantify intraclass correlation: Modifications of Pearson’s product-moment correlation coefficient. Comparisons based on the generalized estimating equations generated by mixed-effect models.	Analysis of data using a mixed-effect linear model that can accommodate a dependent variance-covariance structure.
Multiplicity	Controlling the family-wise error rate (Tukey, Bonferroni, Scheffe and other). Approach to controlling the false discovery rate used in biomarker studies: Benjamini and Hochber.	Analysis of data using a methodology that controls the family-wise error rate.
Multiple Clinical Endpoints	The selection of a single primary endpoint for formal statistical inference, considering that the endpoints are possibly biologically related and positively correlated. Creating a univariate outcome by combining multiple clinical endpoints (weighted measures taking into account the relevance of each endpoint). To compare the two samples based on the endpoint of highest priority first, and, if no winner can be determined, would one move to the endpoint of the next highest priority.	Analysis of data by prioritizing the relevant endpoints or by using a composite endpoint.
Selection bias	To adjust for age, stage, treatment and so forth. Matched samples.	Analysis of data using a multivariate model to simultaneously adjust for confounders Obtention of matched samples. Propensity score weighted.
Publication bias	To publish positive and negative results.	Encourage the objective assessment of molecular signatures by reporting both positive and negative outcomes. Make data publicly available after publication.

## Data Availability

Copies of searches and data extraction sheets will be made publicly available on the online platform Zenodo (zenodo.org) as part of the database collection for all scoping reviews conducted in the PERMIT project.

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
