# Peer review of "Methods for Stratification and Validation Cohorts: A Scoping Review"

_jpm, 2022, doi:10.3390/jpm12050688_

Round 1

Reviewer 1 Report

The paper presents a comprehensive overview of the evidence about the design, building and management of stratification and validation cohorts, emphasizing the need to develop standards and harmonized guidelines to improve the reproducibility in personalized medicine studies.

The topic is very interesting and of great importance in the era of precision medicine and it is presented in an exhaustive and complete way, with the known limitation of considering previous reviews in this scoping review. Arguably, a literature update would have been appreciated.

Before publication, I suggest these minor revisions:

  • Please, modify Figure 1 (PRISMA flowchart describing process for article selection), to make the results complete, clear, and legible.
  • Please verify that Table S6 are introduced in the submission process and are accessible.

Regarding References:

  • In the text, reference numbers should be placed before the punctuation, for example Line 42 “[1,2].”, instead of “.[1,2]”. This correction should be applied throughout the text.
  • Please, correct reference numbers reported in the brackets in line 262: “17, 21, 23, 32-33, 36-38, 40, 42-45, 47, 49-58]”, instead of “[17, 21, 23, 32-33, 36-38, 40, 42-45, 47 49-58]”.
  • Please, check that the paragraph 3.5 does not need a citation.
  • Please reorder reference numbers in line 358: “[54, 62]”. Instead of “[62,54]”.
  • In the paragraph 3.7, eleven citations are introduced, but the reference number [65] is not reported in the following subparagraph. Conversely, in the text is reported reference number [64] not present in lines 362-363. Please check these reference numbers.

Author Response

[REVIEWER'S COMMENT] The paper presents a comprehensive overview of the evidence about the design, building and management of stratification and validation cohorts, emphasizing the need to develop standards and harmonized guidelines to improve the reproducibility in personalized medicine studies.

The topic is very interesting and of great importance in the era of precision medicine and it is presented in an exhaustive and complete way, with the known limitation of considering previous reviews in this scoping review. Arguably, a literature update would have been appreciated.

[ANSWER] Thank you very much for your positive assessment of our work. You are correct that the review may be somewhat outdated (hopefully very little) due to delays beyond our control and the publishing rush over the past months potentially related to our work. In any case, we hope that our work will have an impact, even if there is a slight delay from completion to publication.

[REVIEWER'S COMMENT]

Before publication, I suggest these minor revisions:

  • Please, modify Figure 1 (PRISMA flowchart describing process for article selection), to make the results complete, clear, and legible.

    [ANSWER] Thank you for the comment. We have included some more information in the flowchart to be better understood.

  • Please verify that Table S6 are introduced in the submission process and are accessible.

[ANSWER] Thanks to your comment, we have detected that the table names from S2 to S6 were wrong. They are now fixed.

[REVIEWER'S COMMENT]

Regarding References:

  • In the text, reference numbers should be placed before the punctuation, for example Line 42 “[1,2].”, instead of “.[1,2]”. This correction should be applied throughout the text.

  • Please, correct reference numbers reported in the brackets in line 262: “17, 21, 23, 32-33, 36-38, 40, 42-45, 47, 49-58]”, instead of “[17, 21, 23, 32-33, 36-38, 40, 42-45, 47 49-58]”.

  • Please, check that the paragraph 3.5 does not need a citation.

  • Please reorder reference numbers in line 358: “[54, 62]”. Instead of “[62,54]”.

  • In the paragraph 3.7, eleven citations are introduced, but the reference number [65] is not reported in the following subparagraph. Conversely, in the text is reported reference number [64] not present in lines 362-363. Please check these reference numbers.

[ANSWER] Thank you very much for taking the time to check the references. We have fixed everything you have told us.

Reviewer 2 Report

  1. Figure 1 is needed to replace because some words are missing

Author Response

Thank you for the comment. We have included some more information in the flowchart to be better understood.

Reviewer 3 Report

The article is enjoyable and thought-provoking to read. I think it will contribute t the field.

Author Response

We are very grateful and happy that our work is so well appreciated by the reviewer. Thank you very much.